

# Playing by the rules? Phenotypic adaptation to temperate environments in an American marsupial

Sergio F. Nigenda-Morales[1,2], Ryan J. Harrigan[3] and Robert K. Wayne[1]

[1] Department of Ecology and Evolutionary Biology, University of California, Los Angeles, CA, United States of America
[2] National Laboratory of Genomics for Biodiversity, Center for Research and Advanced Studies, Irapuato, Guanajuato, Mexico
[3] Center for Tropical Research, Institute of the Environment and Sustainability, University of California, Los Angeles, CA, United States of America

Corresponding author
Sergio F. Nigenda-Morales,
snigenda@ucla.edu

## ABSTRACT

Phenotypic variation along environmental gradients can provide evidence suggesting local adaptation has shaped observed morphological disparities. These differences, in traits such as body and extremity size, as well as skin and coat pigmentation, may affect the overall fitness of individuals in their environments. The Virginia opossum (*Didelphis virginiana*) is a marsupial that shows phenotypic variation across its range, one that has recently expanded into temperate environments. It is unknown, however, whether the variation observed in the species fits adaptive ecogeographic patterns, or if phenotypic change is associated with any environmental factors. Using phenotypic measurements of over 300 museum specimens of Virginia opossum, collected throughout its distribution range, we applied regression analysis to determine if phenotypes change along a latitudinal gradient. Then, using predictors from remote-sensing databases and a random forest algorithm, we tested environmental models to find the most important variables driving the phenotypic variation. We found that despite the recent expansion into temperate environments, the phenotypic variation in the Virginia opossum follows a latitudinal gradient fitting three adaptive ecogeographic patterns codified under Bergmann's, Allen's and Gloger's rules. Temperature seasonality was an important predictor of body size variation, with larger opossums occurring at high latitudes with more seasonal environments. Annual mean temperature predicted important variation in extremity size, with smaller extremities found in northern populations. Finally, we found that precipitation and temperature seasonality as well as low temperatures were strong environmental predictors of skin and coat pigmentation variation; darker opossums are distributed at low latitudes in warmer environments with higher precipitation seasonality. These results indicate that the adaptive mechanisms underlying the variation in body size, extremity size and pigmentation are related to the resource seasonality, heat conservation, and pathogen-resistance hypotheses, respectively. Our findings suggest that marsupials may be highly susceptible to environmental changes, and in the case of the Virginia opossum, the drastic phenotypic evolution in northern populations may have arisen rapidly, facilitating the colonization of seasonal and colder habitats of temperate North America.

## INTRODUCTION

Clinal geographic variation can arise as local adaptation within environmental gradients across the geographic range of a species, and can contribute to phenotypic divergence among populations (*Mayr, 1956*; *Endler, 1977*). Some of the most conspicuous traits capable of such responses to changes in the environment are body size, coloration, and body extremity dimensions, especially in species with large geographic ranges (*Millien et al., 2006*). These traits are functionally important, as they can affect numerous physiological and ecological processes in animals (*Caro, 2005*; *Lomolino & Perault, 2007*; *Tattersall et al., 2012*). It is well-known that geographic variation of these traits between populations may follow general ecogeographic patterns; these include Bergmann's (larger body sizes in high, colder latitudes; *Bergmann, 1847*), Allen's (shorter extremities in higher latitudes; *Allen, 1877*) and Gloger's (less pigmentation in high latitudes; *Gloger, 1833*) rules. These patterns are thought to be the result of adaptations to selective pressures imposed by gradients in environmental variables (*Mayr, 1956*; *Millien et al., 2006*) and several hypotheses have been proposed to explain them. For example, thermoregulation, primary productivity and resource seasonality-fasting endurance have all been advanced as explanations for body size variation (*Bergmann, 1847*; *Rosenzweig, 1968*; *Boyce, 1979*; *Lindstedt & Boyce, 1985*). Similarly, heat conservation may explain size change in extremities (*Allen, 1877*) and concealment, thermoregulation, prevention of cold injury and pathogen resistance may account for skin and coat pigmentation variation (*Post, Daniels & Binford, 1975*; *Mackintosh, 2001*; *Caro, 2005*).

However, during or after a process of population expansion into new environments, the ecogeographic patterns may not be observed, possibly because not enough time has elapsed for adaptive changes to occur, or because trait plasticity may be more likely to drive phenotypic differences during expansion (*Ghalambor et al., 2007*; *Pfennig et al., 2010*). Nevertheless, if evidence for phenotypic adaptations is found, traits likely evolved quickly to match these new environmental regimes (*Hairston et al., 2005*; *Bradshaw & Holzapfel, 2006*). Finding the environmental variables associated with geographic variation is important to elucidate the evolutionary processes and mechanisms underlying phenotypic change (*Kamilar et al., 2012*).

The ecogeographic patterns mentioned above broadly apply to mammals (*Ashton, Tracy & De Queiroz, 2000*; *Meiri & Dayan, 2003*), but limited studies have been done in marsupials (*Yom-Tov & Nix, 1986*; *Lindenmayer et al., 1995*; *Quin, Smith & Norton, 1996*; *Cooper, 1998*), and to our knowledge no studies have explored the effect of environmental variables on phenotypic patterns in American marsupials. Marsupials represent the ancestor group of Eutherian mammals; they have lower metabolism and body temperatures than Eutherians, probably making them more susceptible to selective pressures related to environmental fluctuation (*McNab, 1978*; *Tyndale-Biscoe, 2005*).

The Virginia opossum (*Didelphis virginiana* Kerr, 1792) is a nocturnal marsupial widely distributed from northwestern Costa Rica to southern Ontario and British Columbia in Canada. The species likely originated in tropical Central America (*Gardner, 1973*; *Jansa, Barker & Voss, 2014*), and the fossil record and paleoclimate data suggest it recently expanded its range into the seasonal temperate habitats of North America, during the last 15–11 kyr (thousand years) (*Graham et al., 1996*; *Bartlein et al., 1998*; *Morgan, 2008*; *Graham & Lundelius, 2010*). Although widely distributed, the Virginia opossum is mostly absent in xeric environments, and habitats with extremely low temperatures (*Gardner & Sunquist, 2003*). This species has poor thermoregulatory capabilities at low temperatures due to its high thermo-neutral temperature (*Lustick & Lustick, 1972*). However, individuals in northern populations develop a higher fur density during the colder months (*Gardner, 1973*), which might be an adaptation to low temperatures in seasonal habitats. These observations, combined with the tropical origin of the species and its recent expansion into temperate climates, indicate the species is sensitive to low temperatures and adaptations to these new environments may have evolved over a short period of time.

The Virginia opossum shows phenotypic variation across its range for some body dimension and coloration traits. Southern opossum populations have lower body weights and longer tails, (although not necessarily shorter bodies) than those in northern localities (*Gardner & Sunquist, 2003*). In addition, the proportion of the skin depigmented on the naked ears, feet, and tails is greater in northern populations, which also have lighter pelage coloration of the face and the dorsal part of the torso (*Gardner, 1973*; *Gardner & Sunquist, 2003*). These phenotypic characteristics may have been generated as a plastic response to the expansion into colder climates or by rapid adaptive evolution in those environments. Previous research has only generally described patterns of phenotypic variation across the distribution range of the Virginia opossum, and had not considered associations with environmental variables (*Gardner, 1973*; *Koch, 1986*; *Gardner & Sunquist, 2003*). Consequently, the evolutionary processes driving the extensive geographic variation in this species and other marsupials are poorly understood.

Due to its tropical origin, recent colonization of temperate regions, sensitivity to environmental variables and phenotypic geographic variation, the Virginia opossum is a good model species to study the evolutionary processes that have shaped phenotypic variation in marsupial mammals. Here we test if the phenotypic variation observed in the Virginia opossum follows ecogeographic patterns and attempt to determine the adaptive mechanisms that may be driving this variation. If no evidence of ecogeographic patterns are found, it would suggest population structure or plastic responses may have played a greater role in driving the phenotype diversity in the species. We implemented a machine learning algorithm to assess the association of geographic variation with environmental variables and test different adaptive hypotheses. We used environmental predictors corresponding to a wide range of habitat features, including temperature, precipitation, elevation, moisture, and vegetation coverage, that have been suggested to affect coloration, body and extremity size variation (*Lindsay, 1987*; *Ashton, Tracy & De Queiroz, 2000*; *Chaplin, 2004*; *Caro, 2005*; *Kamilar et al., 2012*; *Terada, Tatsuzawa & Takashi, 2012*). We expect different sets of environmental variables to be important predictors of geographic variation depending

on the adaptive hypotheses explaining this variation. For example, we predict that if thermoregulation is the major adaptive explanation of body size variation in the Virginia opossum, it can be expected that variables related to low temperatures to be the most important predictors for this trait. In contrast, seasonality variables (for temperature or precipitation) would be more important if resource availability is playing a larger role in the evolution of body size. In the case of extremity size, we predict temperature or temperature changes would be important explaining extremity size if heat-conservation strategies have played a role in the adaptation of opossums to temperate environments. For skin and coat pigmentation traits, we would expect low temperature variables to be important predictors if thermoregulation or cold injury are the selective mechanisms underlying their variation. Conversely, if variables related to vegetation coverage are better predictors of pigmentation, concealment probably has a more important adaptive role.

## MATERIALS & METHODS

### Museum specimens

We examined 352 (163 females and 189 males) study skins of Virginia opossum museum specimens to measure body and extremity dimensions, and 345 (159 females and 186 males) specimens for skin pigmentation and coat coloration analysis (Table S1). The specimens belong to five different natural history collections and were collected over 145 years (1865–2010; Table S2) along the geographic range of the species, from southern Nicaragua to the northern east coast of the United States (US), including some specimens from the western US in California (Fig. 1). We obtained the coordinates for each specimen collecting locality using two different approaches: (1) the coordinates provided by the original collectors; or (2) we located the collecting site provided by the original collector using the Google Earth program and extracted the geographic coordinates. Table S2, provides the geographic coordinates, year and country in which the specimens were collected. To avoid the confounding effects of age, all specimens measured for this study were adults. We determined the age following *Gardner & Sunquist*'s (*2003*) age estimation method based on tooth eruption. For each specimen, we took measurements to identify variation in body size, body extremities size, proportion of pigmented skin of the extremities and, body and facial coat coloration. Twelve phenotypic traits were measured in total, and because some specimens had damaged body parts, not all specimens were measured for all traits (see Table S1 ). The variation in some of the traits is shown in Fig. 2. This study was done following UCLA's Office of Animal Research Oversight protocols #2011-121-02 and #2011-121-03.

### Phenotypic measurements

To quantify variation in body size, we measured body length (head and body; Fig. 2A) and left hindfoot length (heel to tip of middle digit). Hindfoot size is often used as an alternative to represent body size because these traits frequently co-vary (*Suttie & Mitchell, 1983*; *Martin et al., 2013*). Consequently, since these measurements were positively correlated with each other ($r = 0.547$; $P < 0.001$), we used hindfoot length as an additional body size measurement in subsequent analysis, instead of as a measurement of extremity size. For
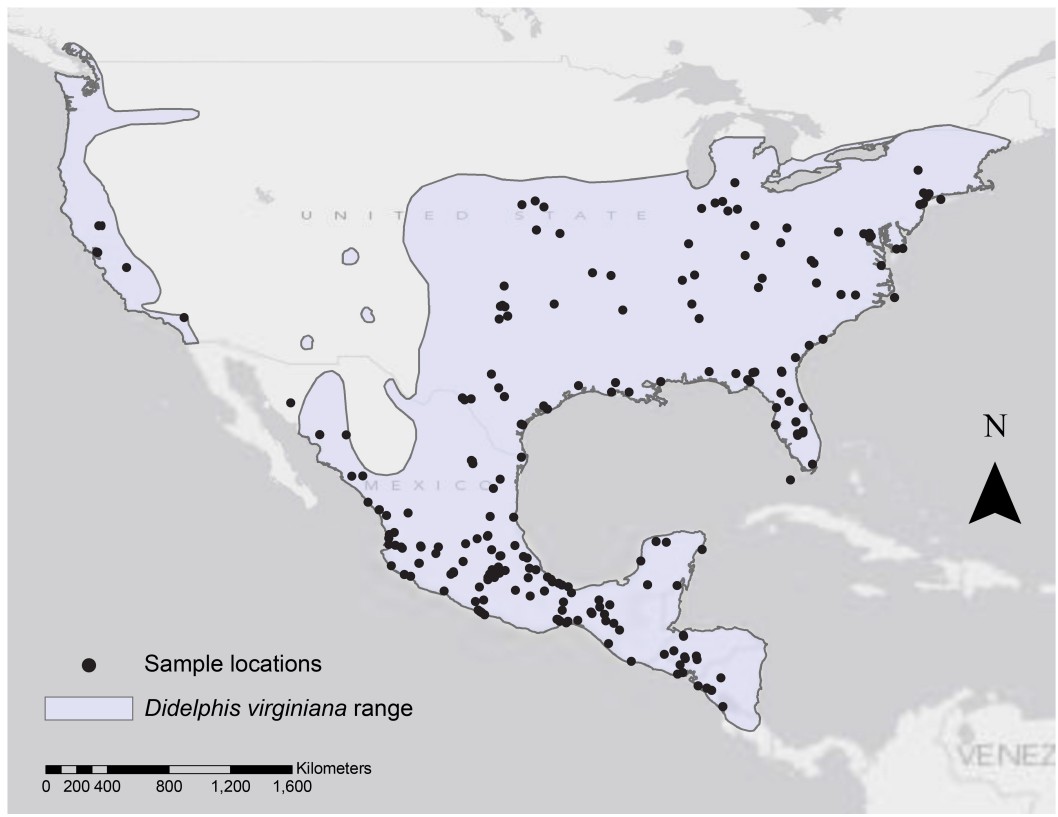

**Figure 1 Geographic range of *Didelphis virginiana* and collecting localities.** The shaded light purple area represents the reported distribution range for the species (*Pérez-Hernandez, Lew & Solari, 2016*) and the collecting localities for the museum specimens analyzed in this study are marked by black dots.

appendage size, we measured the tail length (base to tip; Fig. 2B) and posterior aspect of the ear length (from base to most superior part of the pinna). We measured the posterior aspect of the ear because most specimens had their ears folded and it was impossible to measure the anterior part of the pinnae (the more commonly used measurement). For simplicity, hereafter, we will refer to the posterior aspect of ear length as ear length. All measurements were taken in centimeters to the nearest 0.1 decimal.

We measured four skin pigmentation traits by recording the proportion of the tail, ear, ventral and dorsal aspect of the hindfoot's middle digit that was visibly pigmented (Figs. 2B, 2C). We measured the proportion of pigmentation on both the ventral and dorsal part of the middle digit because populations in the US have hindfeet with light pigmentation on the sole and ventral part of the toes, and in the most northern populations, the light pigmentation reaches the dorsal part of the toes (*Gardner, 1973*).

Body and facial coat coloration measurements were taken using the tristimulus colorimeter Minolta ChromaMeter CR-200 (Minolta, Osaka, Japan). This colorimeter measures the reflectance of a xenon flash light and records the color in the three-dimensional color space CIELab. The lightness axis ($L^\star$) expresses color brightness, with values ranging between 100 and 0 for white and black surfaces, respectively, whereas the $a^\star$ and $b^\star$ axes are

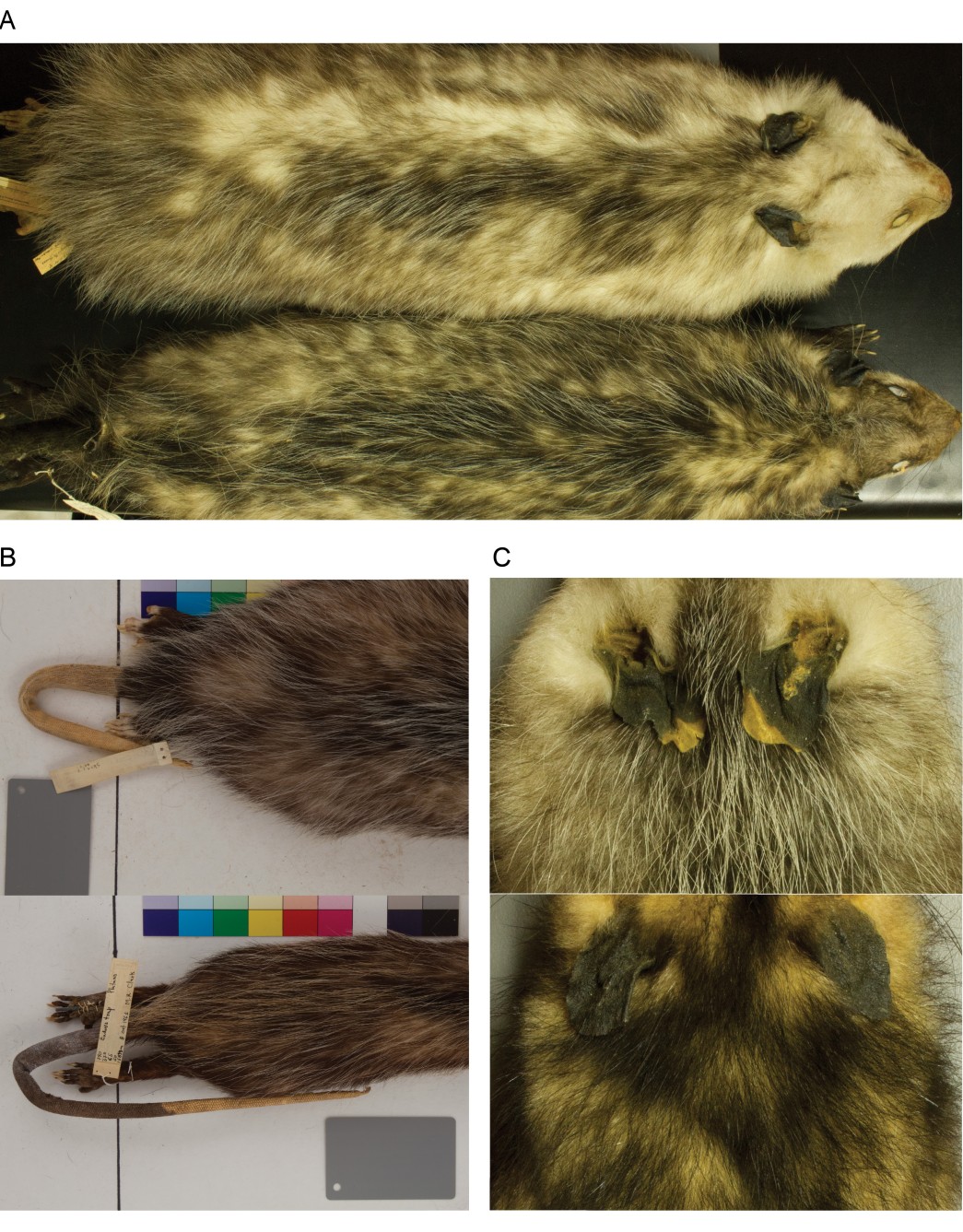

**Figure 2 Phenotypic variation in *Didelphis virginiana*.** Phenotypic differences between specimens from southern and northern populations. (A) Differences in body size (i.e., head and body) and coat coloration of the face and dorsal part of the torso. (B) Variation in tail length, proportion of tail pigmentation, and ventral and dorsal hindfoot toes pigmentation. (C) Dissimilarities in the proportion of ear pigmentation.

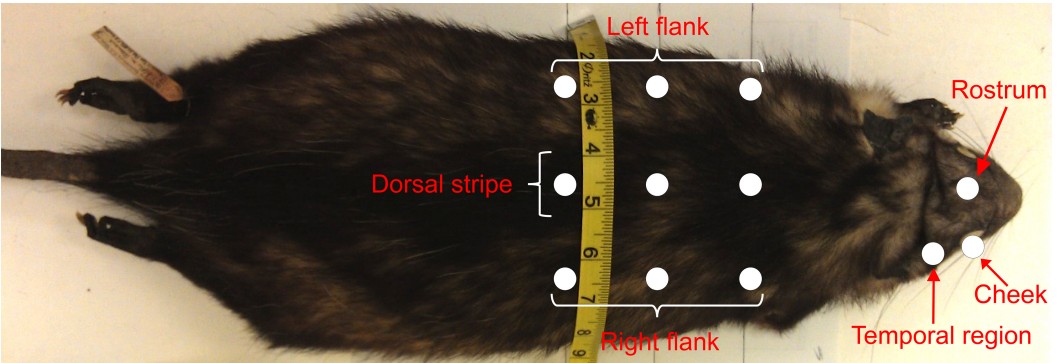

**Figure 3** **Sites on *Didelphis virginiana* museum skin from which reflectance measurements were taken.** Nine sites were measured on the torso (three on each flank and three on the dorsal stripe) and three sites on the face (i.e., rostrum, cheek and temporal regions).

the color coordinates (*Fullerton & Keiding, 1997*; *Clarys et al., 2000*). Since the phenotypes of coat coloration in the Virginia opossum range from dark to light (Fig. 2A) we only used the information of the lightness axis ($L^\star$) in our analysis. The probe with the light source was held at 90° angle to the surface parallel to the body axis, and the reflectance readings were recorded manually. All reflectance measurements in each site were measured in triplicate and averaged to calculate the total lightness value per site. We recorded reflectance values of lightness from three facial traits (rostrum, cheek and temporal region) and from the torso of the body in nine sites (i.e., three on the right flank, three on the dorsal stripe, and three on the left flank; see Fig. 3). The mean of the torso measurements was used as the average torso lightness value.

## Environmental data

For each locality where specimens were collected, we obtained information at 1-km spatial resolution on 12 environmental variables, which have been reported to affect body and extremity size, and pigmentation in mammal species (*Lindsay, 1987*; *Ashton, Tracy & De Queiroz, 2000*; *Chaplin, 2004*; *Caro, 2005*; *Cordero & Epps, 2012*; *Kamilar et al., 2012*; *Terada, Tatsuzawa & Takashi, 2012*). Eight bioclimatic variables related to temperature (annual mean temperature, mean temperature of warmest quarter, mean temperature of coldest quarter), precipitation (annual precipitation, precipitation of coldest quarter), temperature seasonality (mean diurnal range, temperature seasonality) and precipitation seasonality were obtained from the WorldClim database (*Hijmans et al., 2005*; Table S3). These variables maximize variation in North America while minimizing correlation (*Harrigan et al., 2014*). In addition, four variables were extracted from remote-sensing databases: the maximum value of the normalized difference vegetation index (NDVIMAX) that is related to vegetation density and productivity (*Tucker & Sellers, 1986*; *Buermann et al., 2008*); the vegetation continuous field product as an estimate of the percentage of tree coverage (TREECOV; *Hansen et al., 2002*); the monthly backscatter measurements that capture attributes related to surface moisture (ROUGH) (*Long et al., 2001*); and elevation (ELEV) (Table S3 ). Elevation was included in the analyses due to its relationship with

temperature variation, which can influence the variance of several traits we examined (*Blackburn & Ruggiero, 2001*; *Symonds & Tattersall, 2010*; *Kamilar & Bradley, 2011*).

## Data analysis

We carried out Shapiro–Wilk normality and Bartlett homoscedasticity tests for all the traits and analyzed them accordingly. There is important sexual dimorphism in body and appendages size in opossums (*Gardner & Sunquist, 2003*); however, these measurements were normally distributed and a preliminary analysis of our data showed that they have similar patterns of latitudinal variation for males and females (Figs. S1A–S1D). To insure adequate sample size for body and extremity size measurements, we controlled for the effect of sex on these traits, and used the residuals to analyze both sexes together. To test if there were differences in skin pigmentation and coat coloration traits between males and females we used analysis of variance (ANOVA) or Wilcoxon rank-sum test depending on the normality of the data. All traits had homogeneity of variance, but only cheek and torso lightness were normally distributed and were the only coloration traits statistically different between males and females (cheek lightness: $F = 29.85$, $P < 0.001$; coat lightness: $F = 8.85$, $P = 0.003$), therefore we analyzed them separately for both sexes. All tests were done using their standard functions in the R statistical framework version 3.1.3 (*R Core Team, 2015*).

## Tests of correlation with latitude

To determine the relationship between the phenotypic traits and latitude, we carried out Pearson ($r$) or Spearman ($r_s$) correlation analyses depending on the normality of the data in the R statistical framework (*R Core Team, 2015*). Additionally, using the ggplot2 package in R (*Wickham, 2009*), we plotted the trait values against latitude, implementing the non-linear regression loess function to graphically show the trend of correlations and the latitudes for which there is a change in this trend.

## Association with environmental variables

To detect if spatial autocorrelation was present in our phenotypic data, we calculated Moran's I weighted by the Euclidean distance between two points for each trait using the ape package v. 4.1 (*Paradis, Claude & Strimmer, 2004*) in R. To identify the best environmental models for predicting the phenotypic variation in the Virginia opossum, we ran random forest analyses using the randomForest package v. 4.6-12 in R (*Liaw & Wiener, 2002*; *Prasad, Iverson & Liaw, 2006*). Considering measurements from museum specimens that were collected in the same geographic location could contribute to increase spatial autocorrelation in our data; therefore, we averaged the phenotypic values of those specimens (Dataset S1) for the random forest analyses. To run the random forest analyses, we used the values of each phenotypic trait as response variables, and the 12 environmental variables and two geographic variables (i.e., latitude and longitude; Table S3) as predictors. The two geographic variables were included because incorporating geographic information in random forest models allows an evaluation of how much variation in response is explained by environmental variation as compared to simply geographic proximity (*Evans et al., 2011*; *Mascaro et al., 2014*). Decision trees (regression or classification) and random forest methods have no *a priori* assumptions about the relationship between predictor

and response variables, allowing for the possibility to analyze non-linear relationships with complex interactions (*Breiman, 2001*; *Cutler et al., 2007*; *Strobl, Malley & Tutz, 2009*; *Evans et al., 2011*). Random forests analyses are iterations of large number of decision trees, which recursively partition the data into binary homogeneous groups splitting the response variable by the predictor variable explaining most of the remaining variance. The amount of variation in the response variable explained by each predictor is incorporated in the model. Applying a randomized bootstrapping (bagging) method, random forest analysis uses a subset of both response and predictor variables randomly permuted to construct each regression tree and assess the robustness of the model based on the remaining data not included in the tree. If the accuracy of the model decreases appreciably when a variable is left out of the model, that variable is considered an important predictor of the data (*Breiman, 2001*; *Prasad, Iverson & Liaw, 2006*; *Strobl, Malley & Tutz, 2009*).

We ran 10,000 regression trees for each random forest run, and after each run we eliminated the least important variables in the model and re-ran random forest until we identified the most predictive, least complex models, which were composed of three to five predictors that explained the largest amount of variation for each trait. We compared these combined models (which included environmental and geographic variables) with models composed only by the two geographic variables and models including environmental variables only. This was done to detect the role that geography or environment alone play in explaining phenotypic variation in the opossum. Finally, we also tested for spatial autocorrelation in the residuals of the combined models using the Moran's I statistic to be confident that spatial autocorrelation has not affected the results of those models.

# RESULTS

## Latitudinal variation

### Body and extremities size

We found correlation with latitude for all body dimension measurements (see Table 1; Figs. 4A–4D). The correlations with latitude for body size and hindfoot length were positive ($r = 0.314$, $P < 0.001$ and $r = 0.284$, $P < 0.001$, respectively), increasing after 27°N (Figs. 4A, 4B). In contrast, the correlations for tail length and ear length were negative (see Table 1). Tail length, was smaller above 27°N, whereas for ear length smaller values occur only in latitudes above 37°N (Figs. 4C, 4D). In general, opossums with smaller bodies but larger tails were distributed in latitudes below 27°N, and larger individuals with shorter tails were found in higher latitudes. This pattern of variation follows both Bergmann's and Allen's rule for body and extremity size, respectively.

### Skin and coat pigmentation

There was negative correlation between all skin pigmentation traits and latitude (see Table 1; Figs. 4E–4H). The correlation was strong for tail pigmentation ($r_s = -0.701$, $P < 0.0001$) and moderate for toe's ventral pigmentation ($r_s = -0.583$, $P < 0.0001$). The proportion of skin pigmentation was lower above 26°N for all traits (Figs. 4E–4H). For facial coat coloration traits, lightness had a positive correlation with latitude (see Table 1), particularly noticeable for the rostrum ($r_s = 0.679$, $P < 0.0001$) and temporal regions

**Table 1** Pearson and Spearman correlation between *Didelphis virginiana* phenotypic traits and latitude.

| Trait | Sample size | Pearson correlation | Spearman correlation |
|---|---|---|---|
| Body length | 348 | 0.314[**] | n.a. |
| Hindfoot length | 345 | 0.284[**] | n.a. |
| Tail length | 348 | −0.613[**] | n.a. |
| Ear length | 185 | −0.172[*] | n.a. |
| Tail pigmentation | 340 | n.a. | −0.701[***] |
| Ear pigmentation | 293 | n.a. | −0.559[***] |
| Toe ventral pigmentation | 334 | n.a. | −0.583[***] |
| Toe dorsal pigmentation | 338 | n.a. | −0.548[***] |
| Rostrum lightness | 345 | n.a. | 0.679[***] |
| Temporal region lightness | 345 | n.a. | 0.719[***] |
| Cheek lightness | | | |
| F | 159 | 0.517[***] | n.a. |
| M | 186 | 0.512[***] | n.a. |
| Torso lightness | | | |
| F | 159 | 0.298[**] | n.a. |
| M | 186 | 0.251[**] | n.a. |

Notes.

F, Females; M, Males; n.a, Not applicable.

[*]$P < 0.05$.

[**]$P < 0.001$.

[***]$P < 0.0001$.

($r_s = 0.718$, $P < 0.0001$). These traits were darkly or lightly pigmented below 25–26°N or above 30–31°N, respectively, with a steep cline between these latitudes (Figs. 4I, 4J). This is a common pattern we observed across many body dimension and pigmentation traits where opossums below 27°N or above 31°N are phenotypically similar among them (Fig. 4), suggesting that phenotypic variation is mainly driven by individuals in the range between these latitudes probably due to environmental conditions transitioning from tropical to temperate within this range. Finally, males were lighter than females for cheek ($F = 29.85$, $P < 0.001$) and torso coloration ($F = 8.84$, $P = 0.031$). For both sexes, cheek lightness had larger values above 24°N (Fig. 4K). Weak but significant correlation with latitude was found for torso lightness in both sexes (Table 1), with lightness increasing around 29°N (Fig. 4L). Overall, our skin and coat pigmentation data follow Gloger's rule: opossums have higher proportion of their skin pigmented and darker fur coloration on their face and torso in lower latitudes, whereas less pigmented individuals with lighter face and torso are found at higher latitudes.

## Association with environmental variation

Concordant with our results of latitudinal variation, we found that all phenotypic traits show positive spatial autocorrelation (Table S4) and the most important variable for the models of these traits was latitude, whereas longitude was not consistently among the top variables (Fig. 5). Despite this association between morphology and latitude, environment also helped predict morphological characteristics; models with environmental or geographic

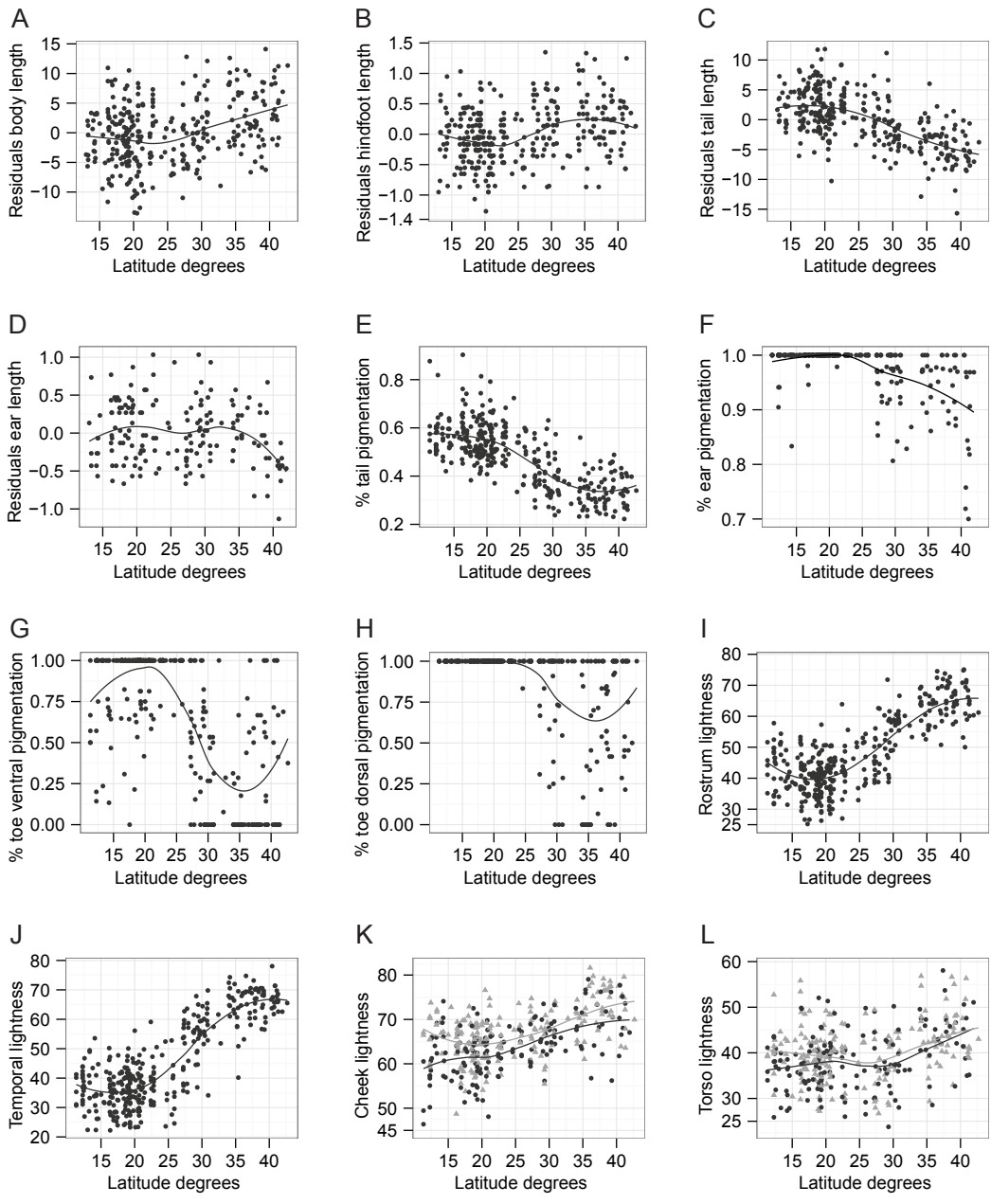

**Figure 4** **Scatter plots of the relationship between phenotypic traits and latitude.** The non-linear loess function line is shown, indicating the trend of the relationship. (A) body length, (B) hindfoot length, (C) tail length, (D) ear length, (E) proportion of tail pigmentation, (F) proportion of ear pigmentation, (G) proportion of toe ventral pigmentation, (H) proportion of toe dorsal pigmentation, (I) rostrum lightness, (J) temporal region lightness, (K) cheek lightness, (L) torso lightness. For cheek and torso lightness, the grey triangles and line represent the data for males while the black circles and line represent the data for females.

**Table 2  Random forest models for each phenotypic trait and the percentage of phenotypic variance they explain.** For most traits, the combined models that include environmental and geographic variables together explain more of the phenotypic variation than the environmental only and geographic only models. When LONG is *in cursive* it means that it was not among the top variables of that particular model (Fig. 5), but was included to take into account spatial autocorrelation. The most important variable of the models is on the left, diminishing in importance towards the right.

| Traits | Best models including environmental and geographic variables. | % of phenotypic variation explained by the models including environmental and geographic variables. | % of phenotypic variation explained by environmental variables only | % of phenotypic variation explained by geographic variables only (i.e., LAT, LONG) |
|---|---|---|---|---|
| Body Dimensions | | | | |
| Body length | LAT, Bio4, Bio11, Bio2, *LONG* | 14.95% | 11.82% | 9.96% |
| Hindfoot length | Bio11, LONG, LAT, Bio4 | 12.75% | 3.39% | 10.77% |
| Tail length | LAT, LONG, Bio4, Bio1, Bio11 | 47.42% | 36.91% | 50.05% |
| Ear length | Bio1, LAT, Bio4, ROUGH, *LONG* | 13.11% | 11.51% | −1.73% |
| Skin Pigmentation | | | | |
| Tail pigmentation | LONG, LAT, Bio4, Bio11, Bio15 | 62.56% | 60.0% | 61.14% |
| Ear pigmentation | LONG, Bio15, Bio4, LAT | 62.18% | 49.07% | 60.78% |
| Toe ventral pigmentation | LAT, Bio 19, LONG, Bio4, Bio11 | 59.86% | 57.57% | 58.9% |
| Toe dorsal pigmentation | LAT, Bio11, Bio15, Bio1, LONG | 31.17% | 27.38% | 34.5% |
| Face coloration | | | | |
| Rostrum lightness | LAT, Bio4, LONG, Bio11, Bio19 | 76.04% | 73.64% | 74.82% |
| Temporal lightness | LONG, LAT, Bio4, Bio11, Bio15 | 78.34% | 72.29% | 79.04% |
| Cheek lightness | | | | |
| F | LAT, Bio4, Bio11, Bio19, *LONG* | 13.75% | 5.74% | 16.87% |
| M | LAT, Bio11, Bio4, Bio15, *LONG* | 29.75% | 27.65% | 24.06% |
| Torso lightness | | | | |
| F | Bio4, Bio12, LAT, Bio11, *LONG* | 19.88% | 18.45% | 14.76% |
| M | Bio4, Bio11, LAT, Bio12, *LONG* | 14.41% | 15.78% | 0.25% |

**Notes.**
F, Females; M, Males; Bio1, annual mean temperature; Bio2, mean diurnal range; Bio4, temperature seasonality; Bio11, mean temperature of coldest quarter; Bio12, annual precipitation; Bio15, precipitation seasonality; Bio19, precipitation of coldest quarter; ROUGH, surface moisture; LAT, latitude; LONG, longitude.

variables alone explained less of the Virginia opossum phenotypic variation than the combined models including environmental and geographic variables together (Table 2). Further, the residuals of these combined models showed small negative Moran's I values, most of them were slightly significant or not significant (Table S5), which indicates the models are appropriate for our analyses since they control for spatial autocorrelation. As such, we explore and explain these more predictive, combined models below.

### Body and extremity size

Combined models explained 12.75% and 14.95% of the total variation in hindfoot and body length, respectively, and 47.42% of tail length variation (Table 2). Aside from latitude, the most important environmental predictors for both body size and hindfoot length were temperature of the coldest quarter (Bio11) and temperature seasonality (Bio4; Table 2; Figs. 5A, 5B), (*Wigginton & Dobson, 1999*). The temperature of the coldest quarter (Bio11) was also the most important environmental predictor of tail length (Table 2; Fig. 5C),

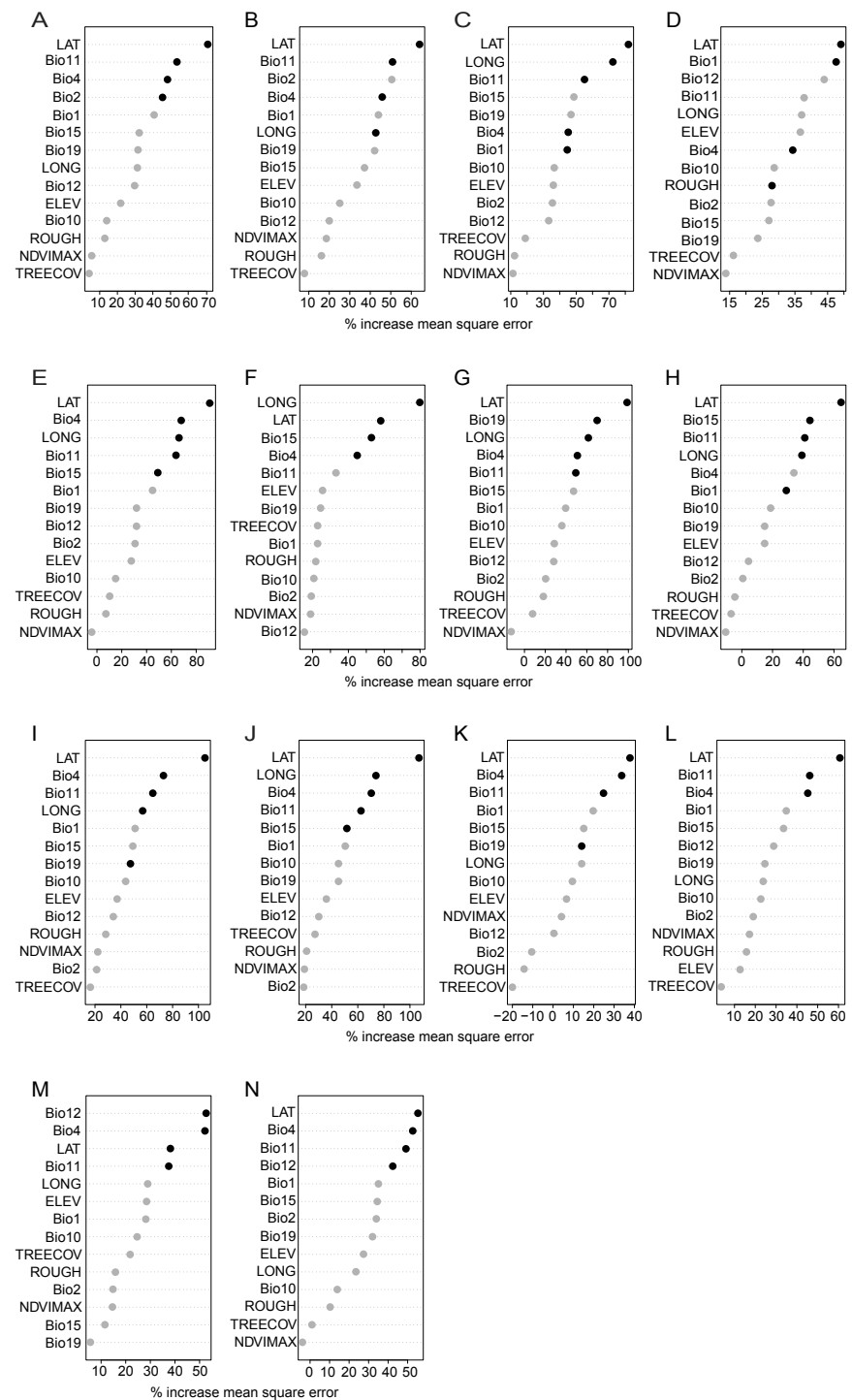

**Figure 5 Importance scores for each predictor variable (including geographic variables) used as input to random forest combined models for all phenotypic traits.** Variables with higher mean square error (calculated as the average increase in squared residuals when the variable is permuted) are more important. Variables shown with a black circle are those that remained *(continued on next page...)*

**Figure 5 (…continued)**
important as the model was refined. (A) body length, (B) hindfoot length, (C) tail length, (D) ear length, (E) proportion of tail pigmentation, (F) proportion of ear pigmentation, (G) proportion of toe ventral pigmentation, (H) proportion of toe dorsal pigmentation, (I) rostrum lightness, (J) temporal region lightness, (K) cheek lightness in females, (L) cheek lightness in males, (M) torso lightness in females, (N) torso lightness in males. Bio1, annual mean temperature; Bio2, mean diurnal range; Bio4, temperature seasonality; Bio10, mean temperature warmest quarter, Bio11, mean temperature of coldest quarter; Bio12, annual precipitation; Bio15, precipitation seasonality; Bio19, precipitation of coldest quarter; NDVIMAX, normalized difference vegetation index maximum value; TREECOV, percent tree cover; ELEV, elevation; ROUGH, surface moisture; LAT, latitude; LONG, longitude.

whereas 13.11% of the ear length variance was explained by the model (and its main environmental predictor, annual mean temperature (Bio1)) (Table 2; Fig. 5D). Together, the distribution of the opossums and the combined models indicate that larger opossums occur (above 27°N) where there is higher temperature seasonality (Fig. 6A). Whereas for the extremity size variation, tail length was reduced above 27°N, in regions where the mean temperature during the winter (i.e., the coldest quarter) is relatively low (Fig. 6B).

### Skin and coat pigmentation

Combined models explained 31.17–62.56% of the variation in skin pigmentation, 13.75–78.34% in face coloration, and 14.41–19.88% in torso coloration (Table 2). The most important and consistent environmental predictors for explaining phenotypic variation in skin pigmentation and face coloration traits were temperature seasonality, temperature of coldest quarter, along with two precipitation variables: precipitation seasonality (Bio15) and precipitation of the coldest quarter (Bio19; Table 2; Figs. 5E–5L). According to the distribution of these predictors and of the opossums, individuals with more depigmented skin and lighter face (distributed above 27°N) were found in localities where temperatures during the coldest quarter (i.e., winter; Fig. 6B) and precipitation seasonality are low (Fig. 6C), while temperature seasonality (Fig. 6A) and precipitation of the coldest quarter are higher (Fig. 6D).

In contrast with other pigmentation traits, the combined models explained relatively less of the variation in torso lightness, for males (14.41%) and females (19.88%), with temperature seasonality, temperature of the coldest quarter and annual precipitation (Bio12) being the most important environmental predictors (Table 2; Figs. 5M, 5N). A pattern of lighter torso coloration was found where temperature seasonality was higher and temperature of the coldest quarter and annual precipitation were lower. These results together with our latitudinal distribution results suggest opossums have high proportion of skin pigmented and darker pelage on the face and torso in humid tropical environments below 26°N, where conditions are warmer, less seasonal and with higher precipitation variability, whereas they are less pigmented towards seasonal, drier and colder habitats above 31°N.

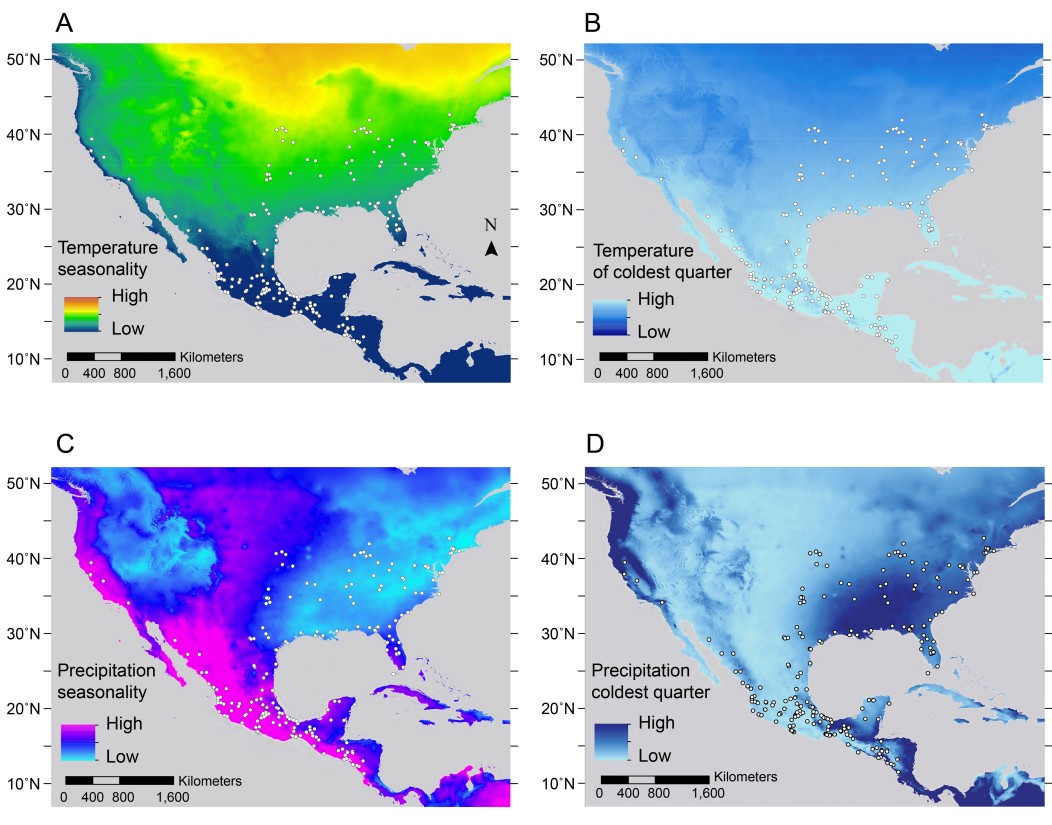

**Figure 6** Maps of geographic variation for the most important environmental predictors associated with trait variation in *Didelphis virginiana*. (A) Temperature seasonality, (B) temperature of the coldest month, (C) precipitation seasonality and (D) precipitation of the coldest quarter. The white dots show the distribution of Virginia opossum specimens used in this study.

## DISCUSSION

### Adaptation to temperate and seasonal environments

Our results indicate that there is a dramatic phenotypic change between tropical and temperate Virginia opossum populations. This change may be driven by natural selection since they adjust to three ecogeographic patterns. The facts that the traits show spatial autocorrelation (Table S4) and that latitude was the most important variable for most of the models (Fig. 5) is not surprising, given the latitudinal gradient we observed in the pattern of phenotypic variation (Table 1) and that environmental variables are also correlated with latitude. Therefore, it is difficult to distinguish the effects of pure spatial autocorrelation when environmental variables that change along similar gradients, which is a limitation of studies perform at large spatial scales. However, the inclusion of latitude and longitude as predictor variables allows us to better tease apart the contributions of geography and ecology separately in our random forest models (*Evans et al., 2011*; *Mascaro et al., 2014*) and control for spatial autocorrelation (Table S5). The results of these models

support the idea that phenotypic variation in the opossum follow ecogeographic patterns that vary with latitude and identify the most important environmental variables responsible for those patterns.

*Koch (1986)* detected a similar trend as we observed, that of increasing body size with latitude using the lower first molar area as body size surrogate, but the association to environmental variables was not tested in that study. The association we found of larger hindfoot and body size with environmental predictors of seasonality (Table 2; Figs. 5A, 5B), is consistent with the resource seasonality (also known as fasting endurance) hypothesis as an explanation for Bergmann's rule. This hypothesis suggests natural selection favors larger individuals in regions of greater seasonality where food availability and energy demands are less predictable (*Boyce, 1979*; *Lindstedt & Boyce, 1985*). For the opossum, one of the most important causes of mortality is starvation due to harsh climate conditions in the winter, especially in the northern part of its distribution (*Kanda & Fuller, 2004*; *Kanda, 2005*). The main factors predicting whether an opossum will survive the winter are body weight and size (*Brocke, 1970*; *Kanda, 2005*). Larger individuals save more energy than smaller ones, owing to a lower metabolism at low temperatures (*McNab, 1978*). Moreover, the seasonality hypothesis predicts that larger individuals accumulate more fat and metabolize it at lower rates than smaller ones, having greater fasting endurance and survival probability (*Lindstedt & Boyce, 1985*; *Millar & Hickling, 1990*). In accordance, under fasting conditions, opossums change from using carbohydrates as energy source to lipid storage (*Weber & O'Connor, 2000*). Our data suggests that the selective pressure of weather-driven food seasonality and availability may have resulted in a phenotypic adaptation of the Virginia opossum towards larger bodies in seasonal and temperate environments above 27°N. This finding is supported by the fact that at least some Australian marsupials conform with Bergmann's pattern in response to temperature variability and food availability (*Yom-Tov & Nix, 1986*; *Quin, Smith & Norton, 1996*).

The variation of body extremities, especially for tail length, follow Allen's rule (Figs. 4C, 4D) and is associated with temperature variables (Table 2; Figs. 5C, 5D). Allen's pattern suggests that natural selection favors individuals with larger body appendages that increase surface area to dissipate heat via conduction in warmer climates, whereas in colder climates shorter appendages would be favored to reduce heat loss (*Millien et al., 2006*; *Tattersall et al., 2012*). This pattern is found in two kangaroo species (*Yom-Tov & Nix, 1986*) while other three Australian marsupials do not show this pattern (*Yom-Tov & Nix, 1986*; *Lindenmayer et al., 1995*; *Cooper, 1998*). Physiological studies have proposed the Virginia opossum is poorly adapted to cold climates due to its high conductivity (i.e., its skin facilitates heat transfer) and low metabolism, and because it relies on behavioral and heat production mechanisms instead of heat conservation as a primary thermoregulatory strategy (*Lustick & Lustick, 1972*; *Hsu, Harder & Lustick, 1988*). However, opossums at high latitudes show heat conservation adaptations to seasonal decrease in temperature by developing a higher pelage density during fall and winter months (*Gardner, 1973*). All physiological studies in this species have used individuals from northern populations above 29°N (i.e., Florida (*McNab, 1978*), Ohio (*Lustick & Lustick, 1972*; *Hsu, Harder & Lustick, 1988*), Michigan (*Brocke, 1970*) and New York *McManus, 1969*), which our results revealed

are phenotypically more similar among them than compared with southern populations. Consequently, further research is needed to explore the physiological characteristics of tropical populations. As we have shown, northern populations may have phenotypic adaptations to colder climates favoring heat conservation compared to populations in the south. It is possible that during the northwards expansion of the species the new selective pressures imposed by colder environments favored individuals with reduced extremities that conserved heat better.

Finally, the conformance to Gloger's pattern of skin and coat pigmentation variation and its association with temperature and precipitation variables (Table 2; Figs. 5E–5N) may be driven by pathogens. The pathogen-resistant hypothesis suggests the higher pathogenic incidence in humid and warm tropical environments (*Guernier, Hochberg & Guégan, 2004*; *Lafferty, 2009*) is the selective pressure driving the increased pigmentation observed in tropical populations, because highly pigmented skin, hair or feathers confer better resistance to pathogenic infection (*Mackintosh, 2001*; *Burtt & Ichida, 2004*). Accumulating evidence suggests that melanocytes, melanosomes and melanin function as integral part of the innate immune system response against invading pathogens (*Mackintosh, 2001*; *Elias, 2007*). For example, darker humans are less prone to bacterial and fungal infections than individuals with light skin (*Mackintosh, 2001*), black feathers on birds are more resistant to bacterial degradation than light ones (*Burtt & Ichida, 2004*), and darker greenfinches (*Carduelis chloris*) have higher survival to protozoan infections than paler ones (*Männiste & Hõrak, 2014*). Similar to the other phenotypic traits we analyzed, the evolution of adaptations in pigmentation (in the opossum's case is depigmentation) must have occurred during the expansion of the species into North America's drier and temperate environments at the end of the last Ice Age.

Taken together, our results suggest that the strong geographic variation observed in body size, extremity size and pigmentation traits in the Virginia opossum represent phenotypic adaptations of a species of tropical origins to more seasonal, colder and drier environments. These adaptations may have arisen rapidly, around 15–11 ka (thousand years ago), during the initial phase of the range expansion of the species (*Graham et al., 1996*; *Morgan, 2008*; *Graham & Lundelius, 2010*), which may have facilitated the survival of individuals and the colonization of temperate North America. This is a feasible scenario since rapid adaptation to environmental changes can be accomplished within a few generations (*Berteaux et al., 2004*; *Hairston et al., 2005*; *Bradshaw & Holzapfel, 2006*).

## Coloration differences in sexes

As torso and cheek coloration were the coat coloration traits with less variation explained by the environmental variables that we analyzed (Table 2), it is possible there are other factors we did not consider in our analysis, for example, predation, sexual selection or communication, that could also be important for explaining their variation. The fact that, for these two traits alone, females were darker than males, suggest sexual selection may be playing a role, and requires further investigation.

### Phenotypic plasticity

There is the possibility that some of the patterns we observed might be achieved via phenotypic plasticity (*Ghalambor et al., 2007*; *Pfennig et al., 2010*). In a marsupial species (*Sminthopsis crassicaudata*), the difference in body size was found to be a response to temperature changes (*Riek & Geiser, 2012*). Our results for body size variation detected low temperatures as an important predictor, making this alternative hypothesis plausible, however, experimental studies should be done to further investigate the role of plasticity in opossum body size variation. Experimental studies in several mammal species (*Ashoub, 1958*; *Harrison, Morton & Weiner, 1959*; *Lee, Chu & Chan, 1969*; *Weaver & Ingram, 1969*; *Serrat, 2013*) have shown that body extremity size of genetically similar individuals (i.e., siblings) varies depending on the temperature at which they are reared, resulting in larger and shorter extremities in warm and cold conditions, respectively. However, the only similar experiment done in a marsupial species did not find differences in extremity size (*Riek & Geiser, 2012*). Finally, Siamese cats (*Iljin & Iljin, 1930*) among other mammals (*Robinson, 1973*) show acrosematic pigmentation, with darker pigmentation on the ears, feet, tail and face, whereas the rest of the body is lighter. This pattern is due to temperature differences in the skin of the appendages and face compared to the core of the body, in which the appendages are the coolest (*Stern, 1968*). Nevertheless, the pigmentation pattern in the Virginia opossum is the opposite of that predicted by acrosematic patterns. The role of plasticity in the Virginia opossum phenotype variation should be further explored, common garden studies rearing individuals with distinct phenotypes in different environmental conditions and obtaining backcrossed generations may allow to assess the heritability of the traits.

## CONCLUSIONS

Due to the lower body temperature and metabolism that marsupials have, compared to Eutherian mammals, they may be more susceptible to the effects of environmental variables and would be under high selective pressure to adaptively respond to environmental changes. For the Virginia opossum, this prediction appears to be true. We have shown that temperature and precipitation variables are important in shaping the geographic variation of body size, extremity size, and skin and coat coloration in this species. Our results contribute to a better understanding of the evolution of phenotypic traits in marsupials and provide evidence that selective pressures from environmental variables influence greatly their phenotypic variation. In the Virginia opossum, variation conforms to three main ecogeographic patterns. The phenotypic divergence observed may have occurred relatively recently, during the last 15 kyr the species has inhabited temperate environments of North America, which shows the ability of this species for expanding in range and rapidly adapting to new conditions. Although we cannot rule out the possibility that adaptive phenotypic plasticity has played some role in driving these phenotypic patterns, our results better support an adaptive response through the recent action of natural selection. Further research on developmental physiology, population structure, demographic history and gene expression would be needed to further test ideas about phenotypic variation in this marsupial species.

## ACKNOWLEDGEMENTS

The authors would like to thank Nicole Corpuz and Adriana Garmendia for their help capturing the phenotypic data. We are thankful with Alfred Gardner and Suzzane Peurach (USNM), Eileen Westwig (AMNH), Chris Conroy (MVZ, University of California, Berkeley), Livia Leon-Paniagua (Museo de Zoología, Facultad de Ciencias, UNAM), Fernando Cervantes and Yolanda Hortelano (CNM, Instituo de Biología, UNAM) for granting us access to their respective museum collections.

### Funding

Funding for this work was provided by the University of California Institute for Mexico and the United States (UCMEXUS Small Grant, ST 11/06-28 and UCMEXUS Dissertation Research Grant, DG-12-77 to Sergio F. Nigenda-Morales), the United States of America National Science Foundation (grant PD-08-1269 to Ryan J. Harrigan), Mexico's Consejo Nacional de Ciencia y Tecnología (UCMEXUS-CONACYT Doctoral Fellowship No. 304742 to Sergio F. Nigenda-Morales) and the University of California, Los Angeles (with scholarship and travel support to Sergio F. Nigenda-Morales). The funders had no role in study design, data collection and analysis, decision to publish, or preparation of the manuscript.

### Grant Disclosures

The following grant information was disclosed by the authors:
UCMEXUS Small Grant: ST 11/06-28.
UCMEXUS Dissertation Research Grant: DG-12-77.
United States of America National Science Foundation: PD-08-1269.
UCMEXUS-CONACYT Doctoral Fellowship: 304742.
University of California.

### Competing Interests

The authors declare there are no competing interests.

### Author Contributions

- Sergio F. Nigenda-Morales conceived and designed the experiments, performed the experiments, analyzed the data, contributed reagents/materials/analysis tools, prepared figures and/or tables, authored or reviewed drafts of the paper, approved the final draft.
- Ryan J. Harrigan conceived and designed the experiments, analyzed the data, contributed reagents/materials/analysis tools, authored or reviewed drafts of the paper, approved the final draft.
- Robert K. Wayne conceived and designed the experiments, contributed reagents/-materials/analysis tools, authored or reviewed drafts of the paper, approved the final draft.

## Animal Ethics

The following information was supplied relating to ethical approvals (i.e., approving body and any reference numbers):

This research was done following UCLA's Office of Animal Research Oversight approved protocols #2011-121-02 and #2011-121-03.

## Data Availability

The raw and processed data for geographic locations, phenotypic measurements and environmental variables used in our study is provided as a Supplemental File.

## Supplemental Information

Supplemental information for this article can be found online at http://dx.doi.org/10.7717/peerj.4512#supplemental-information.

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
