# Peer review of "Playing by the rules? Phenotypic adaptation to temperate environments in an American marsupial"

_PeerJ, doi:10.7717/peerj.4512_

## Round 0.1 · original submission · Major Revisions

I have received two reviews of your paper. Both of them concur in that your manuscript is worth to be published, but it requires more work. I recommend you to make more explicit your hyphoteses and predictions for each of the rules. Please consider to use the latitude as a continuous variable, otherwise you need a better explanation for this decision. Moreover, you should take into consideration the R value used, because it seems not be a high value.

Full consideration of the suggestions of our reviewers will surely improve the quality of your manuscript.

·

Basic reporting

The manuscript ““Playing by the rules? Phenotypic adaptation to temperate environments in an American marsupial” analyses central points in macroecological studies, i.e., the Bergmann, Allen and Gloger rules. The work was performed with a good number of samples and analysed a large area of the distribution of the species Didelphis virginiana. With no doubt this work is original and of scientific relevance, being of interest for macroecologist and biologist. Meanwhile, I have some concerns and suggestions that I consider important. The figs and tables are correct.
Introduction: Authors should explicit their hyphoteses and predictions for each of the rules. I consider this point the weakest of the work. This is a central point, since the choice of the variables will depend on the hyphoteses presented. Authors can not chose a large number of variables and test all of them to see which correlates more with the response variable. I consider that the work would benefit if the authors present the alternative hypotheses for each of the rules and the variables corresponding to the rules. An example of this approach is Torres-Romero et al. (2016): "Bergmann's rule in the oceans? Temperature strongly correlates with global interspecific patterns of body size in marine mammals"
Line 58-60: Authors should give a better explication of the fundaments of each of the rules and the different nivels of analysis: i) intraspecific and (ii) interespecific (assemblage and cross-species).

Experimental design

Authors should make available body size data in some repository or supplemental material for other biologists to perform other studies.
Authors should clarify which variables were used to test which hypothesis
Line 192-195: Why did they not analyse males and females separately?
Line 211: An analysis of spatial regression (SAR sor GLS) is more appropiate for this study. Not taking into account the spatial correlation can lead to increase type I error (look Legendre, 1993,2002; Diniz-Filho et al. 2009).

Diniz-Filho et al. "Blackwell Science, Ltd Spatial autocorrelation and red herrings in geographical ecology"
Legendre, P. (1993) Spatial autocorrelation: trouble or new paradigm? Ecology, 74, 1659–1673.
Legendre, P., Dale, M.R.T., Fortin, M.J., Gurevitch, J., Hohn, M. & Myers, D. (2002) The consequences of spatial structure for design and analysis of ecological field surveys. Ecography, 25, 601–615.

Validity of the findings

From the results, the conclusions are appropriately. Meanwhile, analysis taking into account the spatial autocorrelation are necessary.
Minor consideration
Line 346-347: “Our data suggests Allen’s rule also applies to American marsupials”. The study is intraspecific, the authors can not make this generalization.

·

Basic reporting

If I am honest, I enjoyed reading this manuscript. Reading was like a story, easy to follow, clear and objective. Language and writing style are clear throughout the manuscript, making it easy to read. Information provided in the Introduction and M&M sections is sufficient and necessary to follow the study aims, and they cite relevant to the study, and both new and classic papers are cited. They provided all the raw data and background analyses results in supplementary files. Figures are necessary but not excesive.
Results reporting is appropriate and they are discussed accordingly.

Experimental design

All the items requested in this category are fulfilled. Although the question proposed is not new in biology, it was not answered for the study organism before and it is clearly stated in the introduction. The study absolutely follows high technical and ethical standards. Methodology is clear, even the statistics are well explained, and easily replicable.

Validity of the findings

This is a “clean”, “tidy” study. Data is actually robust, statistically sound, and controlled.
Conclusions are well stated, linked to original research question and limited to supporting results, although some hypotheses are suggested to be tested to complete the study findings.

Additional comments

I think this is the first time it happens to me that I enjoy a manuscript when reviewing from the beginning to the end. I believe this is a good contribution for the journal and for life sciences, and I can only suggest some minor modifications. The design is simple and clear, sampling is wide enough, methods are appropriate, and findings are discussed accordingly. However, I would like to ask the authors why they used grouped latitude instead of using it as the continuous variable it is. I can see they have a break at some point of the distribution, but that should be evidenced using linear models too (see minor comments about it). Also, you used hindfoot length as a proxy for body size, after a R = 0.54. That is not a “high correlation”, and I suggest you to keep body size as size variable.

Minor comments:

L150- I am sorry to say I cannot recall any author in order to make a strong statement, but I am quite sure that an R value of near .500 is rather a moderate relationship instead of “high”. I suggest you to maintain body length as a measurement of size.
L206- Is there a reason to make these groups? If that so, please explain why you need to form groups (although arbitrary, from my point of view). Otherwise, I suggest you to perform linear (lm function in R) models which allow you to include categorical and continuous variables in the same analysis.
L210- Even though we all know the main core packages in R, I suggest to cite them.
L235- Did you check for correlation between environmental variables and latitude/longitude? If so, please mention it in the text
L278- In the M&M section you mentioned that as body size and hindfoot length were correlated you decided to use the later as a proxy for body size, but here you analyzed both. Can you please clarify either the methodology or the results?
L335-337. This sentence is a bit disconnected, I suggest to move it after 339 sentence.
L346-347- This sentence is redundant, you may delete it for the sake of brevity.
L377- Independently?
L392- I do not see the relationship between cheeks color as a diagnosis character and sexual selection of this trait.
Fig. 6A- May be you can rename the variable as “thermal amplitude”, as temperature range high/low looks confusing in the figure.

---

## Round 0.2 · Minor Revisions

I find it very important to take into account the point indicated by the reviewer. Could you show in some way that your results are not affected by spatial autocorrelation?

·

Basic reporting

The authors have made a great effort to incorporate most of the criticisms and suggestions. The work " Playing by the rules? Phenotypic adaptation to temperate environments in an American marsupial ", presents relevant results for the hypotheses presented in the introduction.

Experimental design

Methodologically, only one point worries me, the spatial autocorrelation. The method used is not the most appropriate, since its response variables have spatial autocorrelation, so the residuals of the models will have spatial autocorrelation. If they want to keep the RandomForest method they can follow the approach used by Stuarth-Smith et al. (2013). Stuarth-Smith et al.(2013) uses random forest, but to see if autocorrelation affects his results he did the following: (i) determine how far the spatial autocorrelation decreases and (ii) use only points with distances greater than this distance in the models.

Validity of the findings

The work is novel, important for the understanding of the ecogeographic rules, being an important contribution to the knowledge of the forces that affect in the phenotypic variations along the environmental gradient.

---

## Round 0.3 · Minor Revisions

I apologize for the delay in my decision regarding your manuscript.

We received a minor revision suggestion from Reviewer 1 which you should address,

I am inclined to accept your submission but I have, however, some further concerns that need your attention.

In relation to your predictions, I find them rather circular. As far as I understand, you predict that “if body size variation functions as an adaptive thermoregulatory strategy in the Virginia opossum, variables related to low temperatures should be important predictors for this trait”, which is the same as saying that if body size varies with temperature, variables related to low temperature will predict body size. Different is to say that “if body size variation functions as an adaptive thermoregulatory strategy in the Virginia opossum can be expected that species living e.g. in colder latitudes will show….” The same applies to your final prediction. You are expecting a correlation between temperature and pigmentations traits if thermoregulation mechanisms underlay them. Are you in fact saying that if pigmentation varies because of thermoragulation processes then it is correlated with temperature?

Your main question seems to be about the existence of ecogeographic patterns as an alternative to population structure or plastic responses driving the phenotype diversity in Didelphis virginiana. Please rephrase the paragraph to avoid any misunderstanding.

·

Basic reporting

The work is adequate and well done. Only, I ask the authors to incorporate an I Moran analysis of the residuals of the model, to be reassured that the model was adequate to deal with spatial autocorrelation.

Experimental design

Is ok.

Validity of the findings

Is ok

---

## Round 0.4 · accepted · Accept

Thank you for taking my suggestions into consideration. Your abstract is particularly clear in relation to the general logic followed in your work.
I am happy to accept your manuscript.